# An evidence-based neuro-symbolic framework for ambiguous image scene classification [*]

**Giulia Murtas**  GIULIA.MURTAS@{SIRRIS.BE, BTH.SE}
*EluciDATA Lab, Sirris, Ravensteinstraat 4, Brussels, Belgium*
*Department of Computer Science, Blekinge Institute of Technology, Sweden*

**Veselka Boeva**  VESELKA.BOEVA@BTH.SE
*Department of Computer Science, Blekinge Institute of Technology, Sweden*

**Elena Tsiporkova**  ELENA.TSIPORKOVA@SIRRIS.BE
*EluciDATA Lab, Sirris, Ravensteinstraat 4, Brussels, Belgium*

**Editors:** Leilani H. Gilpin, Eleonora Giunchiglia, Pascal Hitzler, and Emile van Krieken

## Abstract

In this study, we propose a novel neuro-symbolic approach to deal with the inherent ambiguity in image scene classification, combining the usage of pre-trained deep learning (DL) models with concepts from modal logic and evidence theory. The DL models are used to detect objects and estimate their depth in a set of labeled images. The obtained outputs are employed to form a dataset of instances characterizing the possible classes. Subsequently, a multi-valued mapping is defined between the data instances and the considered images resulting into each image being represented by the set of instances associated with it. The obtained mapping is utilized to infer necessity and possibility conditions of each class, or equivalently its upper (plausibility) and lower (belief) probabilities. Based on these interval evaluations, a rule-based and a score-based classifiers are built. The overall method is explainable and directly interpretable, robust to data scarcity and data imbalance. The presented framework is studied and evaluated on an abandoned bag detection use case.

## 1. Introduction

Image scene classification is an intensively researched domain aiming at categorically classifying scenes from static images captured in diverse scenarios, e.g., a train station, a park. Thanks to the availability of vast image repositories and the advancement of deep learning architectures, many approaches have been developed to deal with such tasks, such as RelTR, a Relation Transformer for Scene Graph Generation, presented in Cong et al. (2023). Despite these advances, scene classification still poses challenges, due to the complexity, variability, and often inherent ambiguity of real-world situations. Performing ambiguous scene classification requires high-quality, diverse, and well-annotated datasets. However, in many real-world use cases, such as in abandoned luggage detection, very few labeled examples may be available due to privacy constraints and the rarity of situations that need to be classified. Developing approaches which are robust to data scarcity is thus paramount.

In this work, we propose a novel neuro-symbolic approach for image scene classification composed of three stages: 1) a given set of labeled images is initially fed to an object

---

[*] This research was partially funded by the Flemish Government through the AI Research Program. Veselka Boeva's research was funded partly by the Knowledge Foundation, Sweden, through the Human-Centered Intelligent Realities (HINTS) Profile Project (contract 20220068).

detection and a depth estimation models, both pre-trained and subsequently, attributes characterizing the input images are derived from the outputs provided by the models, forming a set of instances; 2) a multi-valued mapping between the set of instances and the images to be categorized is constructed, by associating each instance with the set of images in which it appears; 3) each class is described in terms of its necessity and possibility conditions or its plausibility and belief values, through the multi-valued mapping. The first presented classification model (rule-based) exploits decision rules derived by the determined necessity and possibility constraints. The second model (score-based) defines a scoring function based on the measure of plausibility and belief of the classes.

The potential of the proposed framework is demonstrated on the task of abandoned object detection. Detecting abandoned objects is a critical task in video surveillance systems and automated monitoring of public spaces. Traditional approaches for abandoned object detection rely on background subtraction and motion tracking, while recent methods are DL-based. Employing neuro-symbolic AI, which combines low-level perception with high-level reasoning, facilitates the development of explainable, transparent, and trustworthy methods that are also less computationally heavy and data-greedy than purely DL-based approaches. The performed experiments establish the method's ability to derive meaningful decision rules from a limited number of labeled examples, allowing to overcome often-encountered challenges related to the lack of sufficient annotated data. Moreover, as the results of the classifiers are based on decision rules which are directly interpretable, the approach is explainable, a clear advantage over fully black-box solutions, and can be easily updated as more data become available.

## 2. Related work

Inherently ambiguous situations are often encountered in the context of abandoned object detection, and a large volume of data is seldom available due to privacy-related constraints. Traditional methods tackle the task by performing background subtraction with techniques such as Gaussian Mixture Models (Mukherjee et al. (2015)) to identify static objects abandoned by a person who has left the scene. More recently, DL-based methods have been developed, capable of higher flexibility, although requiring large amounts of training data. CNN-based architectures like YOLO (Song et al. (2025)) and R-CNN (Park et al. (2020)) have been exploited to detect abandoned objects in surveillance footage. Traditional and modern paradigms are often combined in current research. In Zhou and Xu (2024), an enhanced abandoned object detection method focusing on small and occluded objects is proposed. The model integrates an adaptive learning rate to perform accurate background subtraction which responds to scene changes in the images and allows to reduce noise in the data. Then, the SAO-YOLO network (Small Abandoned Object YOLO), capable of detecting small and occluded objects, is applied to the images. Aiming at detecting abandoned objects on highways, for traffic safety and accident prevention, Song et al. (2025) combines dynamic background modelling leveraging MOG2, a Gaussian Mixture Model, and YOLOv9, an efficient object detection network. Low-confidence object detections are cross-verified with the Iterative Difference of Background Modelling algorithm, which compares sequential background frames to detect stationary objects, in order to reduce false alarms and missed detections.

Although not specific to the task of abandoned object detection, neuro-symbolic approaches have been introduced in the DL-dominated field of computer vision to mitigate the issues caused by low quality, ambiguous, or scarcely available data, and to obtain explainable models whose decisions can be understood and trusted by users, especially in industrial applications, while also presenting lower computational costs. In Wang et al. (2023), the issue of lack of annotated image data is tackled, especially common in domains such as healthcare, where labeling data requires advanced expertise. The work combines a pre-trained computer vision model which extracts features from the unlabeled images, and an inductive logic learner module inferring logic-based rules that can be exploited for the annotation. A human in the loop is queried to confirm the labeling of uncertain samples and to improve the derived logic-based rules. The study delivers promising results, but the reached accuracy is not yet on par with the labeling of human experts, on which it still relies for feedback in the active learning portion of the method pipeline.

Evidence (Dempster-Shafer) theory and modal logic have been integrated into workflows for image classification tasks to assist DL models in dealing with uncertainty and imprecision in the data. In Zhang et al. (2023), evidence theory is used to re-label the training set, assigning ambiguous images to a *meta-category*, i.e., a subset of all possible categories, by selecting the meta-category with the highest degree of belief for each selected image. Ambiguous images are defined as samples showing features of multiple classes, i.e., situated in the overlap of multiple categories. By re-training on the dataset updated with meta-categories, the model learns without overfitting to incorrect labels or misclassified examples. This approach, however, is not suited for binary classification. In Tian et al. (2024), Dempster-Shafer theory is leveraged to pull together the results of a set of classifiers trained on different versions of an over-sampled dataset. The paper tackles the challenge of dealing with unbalanced datasets and the issues of generating synthetic data, which can lead to increased uncertainty and inconsistency in the new data distribution. In summary, neuro-symbolic methods exploiting modal logic and evidence theory show potential in tackling scene classification tasks involving ambiguity, while compensating for typical shortcomings of DL-based approaches, i.e., data-greediness and lack of interpretability.

## 3. Multi-valued Interpretations of Plausibility and Belief in Modal Logic

This section introduces the concepts from the theory of multi-valued mapping (Aubin and Frankowska, 1990) and the modal logic interpretations of plausibility and belief measures (Boeva et al., 1998; Tsiporkova et al., 2000) leveraged in the proposed approach.

A *multi-valued mapping* $F$ from a set $X$ into a set $Y$ associates to each element $x$ of $X$ a subset $F(x)$ of $Y$. The *domain* of $F$, denoted $\mathrm{dom}(F)$, is defined as $\mathrm{dom}(F) = \{x \mid x \in X \wedge F(x) \neq \emptyset\}$. Given a subset $B$ of $Y$, the *inverse image* and the *superinverse image* of $B$ under $F$ are the subsets $F^-(B)$ and $F^+(B)$ of $X$, respectively:

$$F^-(B) = \{x \mid x \in X \ \wedge \ F(x) \cap B \neq \emptyset\} \qquad F^+(B) = \{x \mid x \in \mathrm{dom}(F) \ \wedge \ F(x) \subseteq B\}.$$

A visualization of the defined inverse and superinverse images is shown in Figure 1.

Evidence theory, or Dempster-Shafer theory, is a generalization of probability theory. Dempster (Dempster, 2008) has shown that a multi-valued mapping $F$ from a set $X$ into a set $Y$ carries a probability measure $P$ defined over subsets of $X$ into a system of upper

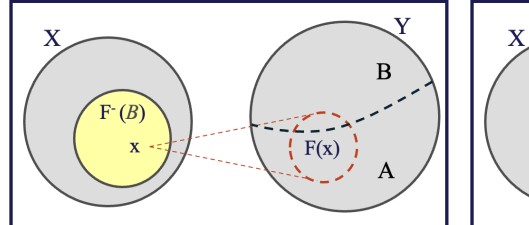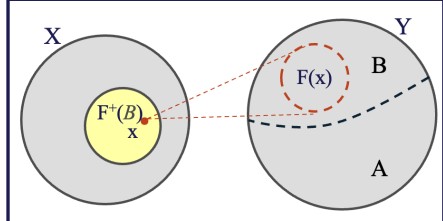

Figure 1: A schematic illustration of the concepts of inverse and superinverse images.

and lower probabilities over subsets of $Y$, where the probability correlated to a set is meant as the probability of each element in the set. Shafer called upper and lower probabilities plausibility and belief functions (Shafer, 1976). $F$ is shown to induce *plausibility* Pl and *belief* Bel functions on $\mathcal{P}(Y)$, as follows:

$$\mathrm{Pl}(B) = P(F^-(B) \mid \mathrm{dom}(F)) \qquad \mathrm{Bel}(B) = P(F^+(B) \mid \mathrm{dom}(F)). \qquad (1)$$

Pl and Bel are only well defined if $P(\mathrm{dom}(F)) > 0$. Note that plausibility and belief measures come in dual pairs. For any belief measure Bel on $\mathcal{P}(Y)$, the $\mathcal{P}(Y) \to [0,1]$ mapping Pl defined by $\mathrm{Pl}(A) = 1 - \mathrm{Bel}(\mathrm{co}\,A)$, where A is a subset of Y as seen in Figure 1, is a plausibility measure on $\mathcal{P}(Y)$.

Modal logic has been developed to present arguments involving the notions of *necessity* $\Box$ and *possibility* $\Diamond$ (Chellas, 1980). The ideas presented in this work are based on the semantics of modal logic using the concept of a standard model. A *standard model* of modal logic is defined as a triplet: a set of possible worlds, a binary relation on this set (*accessibility relation*), and the *value assignment function* by which truth or falsity of each atomic proposition in each world is assigned. In this context, *necessary* propositions are those which are true in all possible worlds, whereas *possible* propositions are those which are true in at least one possible world. Worlds are abstract objects that can intuitively be viewed as possible states of affairs, situations, or scenarios. A proposition $p$ is *true in a model* if it is true in each possible world in this model.

As discussed above, evidence theory is closely related to the theory of multi-valued mappings. Furthermore, set-valued interpretations of plausibility and belief measures in modal logic have been proposed in a number of studies, e.g., Boeva et al. (1998), Tsiporkova et al. (2000). Namely, it is shown that a plausibility measure and a belief measure can be expressed in terms of conditional probabilities of truth sets of possibilities and necessities.

## 4. Methodology

The proposed neuro-symbolic framework combines two different modelling paradigms, based respectively on DL and on modal logic. Namely, the outputs from two pre-trained DL models are initially exploited to extract properties from the images of interest, and subsequently, set-valued interpretations of plausibility and belief measures are employed to build classifiers based on the extracted attributes. The schematic workflow of the proposed approach is shown in Figure 2. The single steps are detailed in the following sections.

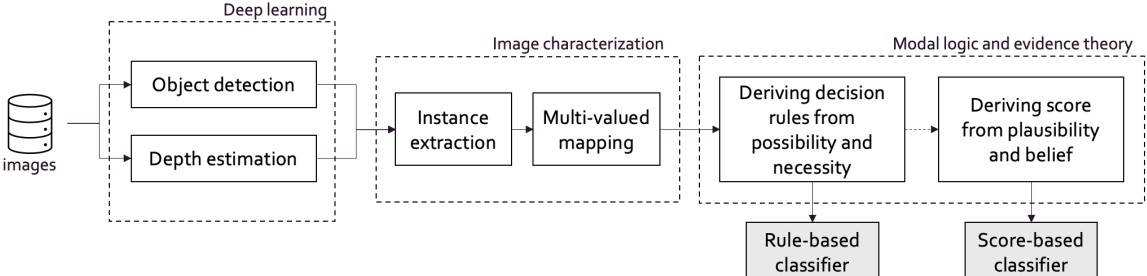

Figure 2: A schematic illustration of the workflow of the proposed approach.

## 4.1. Problem formulation

A binary image scene classification task is considered. Given a set of positively- and negatively-labeled images, the proposed framework aims to characterize the two classes by extracting attributes (also called instances) and to build a classification model learning a mapping from the extracted attributes to the set of possible labels. The obtained model will then be able to classify unseen images based on their relevant instances.

In the above context, a multi-valued mapping $F$ as seen in Section 3 can be defined from the set of all extracted instances $\mathcal{X}$ to the set of images $\mathcal{Y}$ that maps each instance $x \in \mathcal{X}$ to a subset of samples $F(x) \subseteq \mathcal{Y}$ in which the property described by the instance is satisfied. The inverse and superinverse images $F^-(\mathcal{Y}^+)$ and $F^+(\mathcal{Y}^+)$ describing the positive class $\mathcal{Y}^+$ can then be built. The same can be done for the negative class $\mathcal{Y}^-$. Here, $\mathcal{Y}^+$ and $\mathcal{Y}^-$ take the place of A and B in Figure 1. $F^+(\mathcal{Y}^+)$ forms the necessity condition for an image (represented as a bag of instances) to be positively recognized, while $F^-(\mathcal{Y}^+)$ presents the possibility condition for it to be labeled as positive. In other words, the super-inverse image of the positive class includes the instances that are mapped to the necessarily positive samples, while the inverse image consists of the instances mapped to samples that have a possibility of belonging to the positive class. The necessity and possibility conditions referring to the positive class $\mathcal{X}^+$ are described by the following expressions:

$$\Box \mathcal{X}^+ = \bigvee_{x_i \in F^+(\mathcal{Y}^+)} x_i \quad \text{and} \quad \Diamond \mathcal{X}^+ = \bigvee_{x_i \in F^-(\mathcal{Y}^+)} x_i. \tag{2}$$

The negative class can be similarly explained in terms of its necessity $\Box \mathcal{X}^-$ and possibility $\Diamond \mathcal{X}^-$ conditions. These representations of the two classes can be used to analyze and evaluate the discriminative potential of the set of instances extracted from the given images. For example, the set $F^+(\mathcal{Y}^+) \cup F^+(\mathcal{Y}^-)$ contains all the instances that contribute to the discrimination between the two classes, while the instances in the set $F^-(\mathcal{Y}^+) \cap F^-(\mathcal{Y}^-)$ are met in both classes. The ratio between the cardinalities of these two sets can be considered as a quantification of the model's discriminatory potential.

## 4.2. Rule-based classifier

The inferred necessity conditions for the two classes can be used to define decision rules and predict the class of unseen images. The proposed rule-based classifier exploits the logic rules below (Eq. 3). Namely, an image must be assigned to the positive (negative) class if

the bag of instances representing it, denoted by $X_i$, satisfies the necessity condition for the positive (negative) class:

$$\text{IF } \Box\mathcal{X}^+(X_i) \text{ THEN } X_i \in \text{positive class} \quad \text{IF } \Box\mathcal{X}^-(X_i) \text{ THEN } X_i \in \text{negative class.} \quad (3)$$

Note that if $X_i$ does not satisfy any of the two decision rules, it is assigned to a "none of known" class. The task can also be considered as a one-class problem by using only one of the two decision rules given in Eq. 3.

### 4.3. Score-based classifier

For each new unseen image, we can compute its plausibility and belief to belong to one of the two classes. The plausibility and belief of $X_i$ to belong to the positive class are computed as the ratio of instances making out, respectively, the possibility and necessity conditions of the two classes, which are satisfied by $X_i$, as defined in Eq. 4 below.

$$\text{Pl}^+(X_i) = |\, \Diamond\mathcal{X}^+(X_i)\,|\,/\,|\, \Diamond\mathcal{X}^+\,| \qquad \text{Bel}^+(X_i) = |\, \Box\mathcal{X}^+(X_i)\,|\,/\,|\, \Box\mathcal{X}^+\,|. \qquad (4)$$

The plausibility $\text{Pl}^-$ and belief $\text{Bel}^-$ of $X_i$ to belong to the negative class can be similarly computed. The so-calculated plausibility and belief can be used to obtain confidence scores associated with the class prediction. Namely, a scoring function $S$ can be defined which combines the four computed variables, producing a value in the interval [0,1] that can be interpreted as the likelihood of $X_i$ to belong to the positive class. Two alternative approaches for constructing $S$ are proposed, see Eq. 5 and Eq. 6 below.

$X_i$ can be represented by the two intervals $[\text{Bel}^+(X_i), \text{Pl}^+(X_i)]$ and $[\text{Bel}^-(X_i), \text{Pl}^-(X_i)]$. The width of the intervals is correlated with the uncertainty associated to $X_i$. Subsequently, inspired by the work of Boeva and De Baets (2004), $S$ can be based on evaluating the degree of overlap between the two intervals:

$$S(X_i) = \begin{cases} 1 & if \, \text{Bel}^+(X_i) \geq \text{Pl}^-(X_i) \\ 0 & if \, \text{Bel}^-(X_i) \geq \text{Pl}^+(X_i) \\ \dfrac{\text{Pl}^+(X_i) - \text{Bel}^-(X_i)}{\left(\text{Pl}^+(X_i) - \text{Bel}^-(X_i)\right) + \left(\text{Pl}^-(X_i) - \text{Bel}^+(X_i)\right)} & otherwise \,. \end{cases} \qquad (5)$$

Alternatively, $S$ can defined as ratio of available evidence supporting the positive class:

$$S(X_i) = \frac{\text{Pl}^+(X_i) + \text{Bel}^+(X_i)}{\left(\text{Pl}^+(X_i) + \text{Bel}^+(X_i)\right) + \left(\text{Pl}^-(X_i) + \text{Bel}^-(X_i)\right)}. \qquad (6)$$

## 5. Experiments and Results Discussion

### 5.1. Datasets

The method is validated on two datasets, PETS2006 and AVS2007, representing realistic abandoned bag scenarios. The PETS2006 dataset contains videos with multi-sensor sequences depicting abandoned luggage scenarios at a train station scene. Static frames are extracted from the videos in order to apply the proposed approach. Ground truth is not available for both the object detection task and the abandoned bag scene classification. The

labels for the latter, i.e., indicating whether the represented scene illustrates an abandoned bag scenario, have been manually identified and created. The considered dataset consists of 1325 images, of which 95% do not contain an abandoned object, while in the remaining 5% an abandoned bag can be detected.

The AVS2007 dataset (Advanced Video and Signal Based Surveillance) provides benchmark datasets for testing and evaluating detection and tracking algorithms. The i-LIDS bag subset of AVS2007 is considered, as it depicts abandoned luggage scenarios. The dataset comprises of 161 images, 14% of which shows an abandoned object. Labels indicating whether an abandoned bag is present in the image have again been manually created.

### 5.2. Object detection and instance extraction

The first phase of the proposed framework, as depicted in Figure 2, consists in feeding the data to pre-trained DL models to detect the objects of interest (in this case, people and bags) and obtain an estimation of their depth in the image. A pre-trained OneFormer model Jain et al. (2023) is first applied on each image, returning the classes of the detected objects together with the coordinates of the bounding boxes indicating where each object can be found in the image. Subsequently, the Depth Anything V2 model Yang et al. (2024) is used to estimate the depth of the detected objects. Depth estimation assists scene understanding as it allows to more accurately locate objects in a 2D image: two objects that may appear close or overlapping might actually turn out to be far away when depth is taken into account. Figure 3 illustrates the type of outputs obtained by the two models.

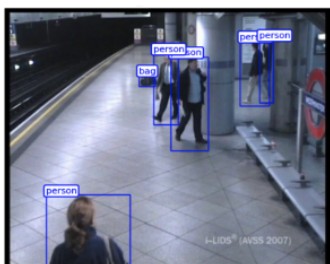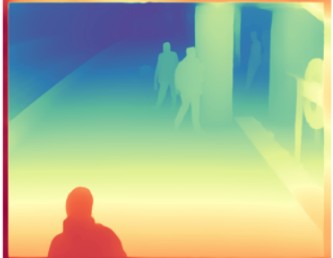

Figure 3: Object detection and depth estimation results

The obtained outputs are exploited to derive meaningful attributes which characterize the images in view of the use case at hand, covering the objects (people and bags) present in the image and the relationships between the bag of interest and the people detected in the image, i.e., the overlap between their bounding boxes and the distance between the bag and the person closest to it. The distance between two objects is intended as the distance between the centers of their bounding boxes while considering the estimated depth of each object, i.e., a 3-dimensional Euclidean distance is calculated. Subsequently, the calculated distances are binned into five overlapping ranges formed by increasing the radius of concentric circles (defined as a ratio of the maximum observed distance in the set, e.g., if min_distance_below_0.1, the distance is lower than $0.1 * maximum\_observed\_distance$) with the bag of interest in their center. A visualization of the distance calculation and of the attributes indicating the possible overlap options among a detected bag and person is

provided in Figure 4. The complete attribute list can be consulted in the first column of Table 1. All extracted attributes are supposed to be binary.

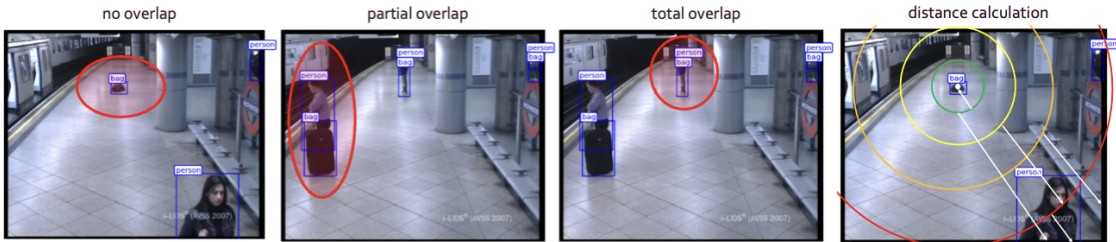

Figure 4: Overlap types and distance calculation for a selected bag

## 5.3. Construction of classifiers

In the two considered datasets, a multi-valued mapping is constructed from each extracted attribute, also called instance in this context, to the subset of images in the training set in which the attribute appears. Subsequently, the inverse and superinverse images of the positive and negative classes under the constructed multi-valued mapping are derived as detailed in Section 4.1. Based on them, the necessity and possibility conditions for the two classes are defined, see Table 1. For example, the inferred necessity of the positive class {*abandoned bag*} and the negative class {*non-abandoned bag*} for the PETS2006 dataset are:

$$\Box\{abandoned\} = min\_distance\_above\_0.5 \vee min\_distance\_above\_0.75$$
$$\Box\{non\text{-}abandoned\} = contains\_person\_but\_no\_bag \vee has\_partial\_overlap$$
$$\vee\ has\_total\_overlap \vee min\_distance\_below\_0.1$$

In this example, the second condition for the abandoned class implies the first, but not vice versa. Eight instances contribute to the discrimination between the two classes in the PETS2006 dataset (one fewer in the AVS2007 dataset, see Table 1). Analogously, the possibility of the two classes can be represented using the data in Table 1. The ambiguity aspect is captured by the instances common to the two classes, indicated below by their index: $ambiguous\_evidence = \Diamond\{abandoned\} \cap \Diamond\{non\text{-}abandoned\} = \{0, 1, 5, 7, 8\}$. Consequently, the possibilities of the two classes can be expressed as shown below. This representation offers an insightful interpretation of the possibility, i.e., there is a possibility that unseen image belongs to one of the two classes when either the necessity condition for this class is satisfied or the image characterization contains at least one of the ambiguous evidence attributes.

$$\Diamond\{abandoned\} = \Box\{abandoned\} \vee ambiguous\_evidence$$

$$\Diamond\{non\text{-}abandoned\} = \Box\{non\text{-}abandoned\} \vee ambiguous\_evidence.$$

Following Eq. 3, the decision rules exploited by the rule-based classifier are defined below:

IF $(9 \vee 10)$ THEN $x \in \{abandoned\}$    IF $(2 \vee 3 \vee 4 \vee 6)$ THEN $x \in \{non\text{-}abandoned\}$

Note that if image $x$ does not satisfy either of the above decision rules it is assigned to the "none of known" class, avoiding a misclassification. A more granular characterization of the

Table 1: Inverse (possibility) and superinverse (necessity) images for the two classes.

| attributes | $possibility_+$ | $possibility_-$ | $necessity_+$ | $necessity_-$ |
|---|---|---|---|---|
| 0: contains_bag | **True** | **True** | False | False |
| 1: contains_person | **True** | **True** | False | False |
| 2: contains_person_but_no_bag | False | **True** | False | **True** |
| 3: has_partial_overlap | **True**$^*$ \| $False$ | **True** | False | False$^*$ \| **True** |
| 4: has_total_overlap | False | **True** | False | **True** |
| 5: has_no_overlap | **True** | **True** | False | False |
| 6: min_distance_below_0.1 | False | **True** | False | **True** |
| 7: min_distance_above_0.1 | **True** | **True** | False | False |
| 8: min_distance_above_0.25 | **True** | **True** | False | False |
| 9: min_distance_above_0.5 | **True** | False | **True** | False |
| 10: min_distance_above_0.75 | **True** | False | **True** | False |

$^*$ These are the values for the AVS2007 data set. All other values are the same for both datasets.

two classes can be obtained by computing their plausibility and belief scores. The values are calculated for each image, using Eq. 4. Subsequently, the scores are used in Eq. 5 or Eq. 6 to compute the likelihood of the image to belong to the class {*abandoned bag*}.

## 5.4. Evaluation and discussion

To evaluate the potential of the proposed method, the built classifiers are run on the two previously discussed datasets. The datasets are randomly split into training and test sets in proportion 80/20, while maintaining the percentage of samples belonging to the two classes fixed. In case of the rule-based classifier, each image is assigned to a class based on which decision rule it satisfies. Images not satisfying either rule are considered uncertain ("none of known" class). The performance of the classifier on the datasets is shown in Table 2. The reported values are the average results over 20 iterations, where the training and test sets were randomly sampled at each iteration. As no sample is misclassified, the significant metrics to report pertain to the percentage of samples considered ambiguous by the model.

Table 2: Performance of the rule-based classifier on the two datasets.

| Metric (%) | AVS2007 | PETS2006 |
|---|---|---|
| Detected positives | 50 | 76.7 |
| Detected negatives | 60.2 | 97.7 |
| Positives in "none of known" | 50 | 23.3 |
| Negatives in "none of known" | 39.8 | 2.3 |
| Overall in "none of known" | 41 | 3.5 |

In case of the score-based classifier, the already computed necessity and possibility conditions are used to obtain the plausibility and belief measures for each class, by applying the formulas in Eq. 4. The two scores defined in Eq. 5 and Eq 6, named Interval-based Abandonment Risk (IAR) and Abandonment Risk (AR) in the context of the explored use case, model the likelihood of an image to depict an abandoned object. The obtained score can be directly taken as reference by, e.g., users working in surveillance, or a threshold can be set in order to trigger an alarm for events requiring intervention. In Figure 5, the

ROC curves of the classifiers on the two datasets are shown, illustrating the variation in the model's performance when varying the threshold. As it can be seen, both IAR and AR perform well on the PETS2006 dataset, However, AR appears to be more robust to data scarcity and complexity, performing better on the AVS2007 dataset, which is, in fact, much smaller than the PETS2006 dataset and contains more complex scenarios.

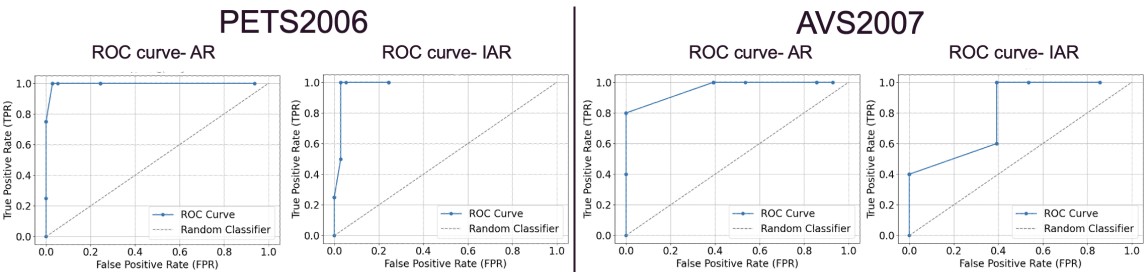

Figure 5: ROC curve for the two datasets when using both IAR and AR scores.

The shown results validate the discrimination potential of the proposed approach in detecting scenes containing abandoned objects. The rule-based classifier allows to avoid misclassifications and signals uncertain scenarios which cannot be assigned to a class due to lack of evidence. The score-based classifier quantifies the risk of an abandoned object being present in a scene. This provides a more detailed representation of the risk level of a given situation, and allows a threshold to be set to trigger alarms in high-risk scenarios. The presented framework leverages the outputs of pre-trained DL models to extract robust logical rules from the defined pertinent attributes characterizing the images, while being significantly less sensitive to data scarcity and imbalance than fully DL-based methods. As the attribute extraction is strictly dependent on the performance of the pre-trained models, it is fundamental to ensure that the models perform sufficiently well on the considered dataset. More specifically, the performance of the object detection DL model poses an upper bound to the performance of the overall framework. If labels for object detection are available in the training set, the ground truth can be exploited when extracting attributes, to avoid adding uncertainty due to incorrect or missed detections to the framework's results.

## 6. Conclusion

This study has introduced a neuro-symbolic framework for handling ambiguity in image scene classification tasks. It combines pre-trained DL models with modal logic interpretations of plausibility and belief functions. It has been evaluated in the context of the abandoned object detection use case. As the method exploits logical rules and derived evidence scores for the sample classification, it is explainable and directly interpretable by design. The uncertainty of classifications is embedded in the process through the distinction of necessity ($\Box$) and possibility ($\Diamond$) in the rule-based classifier, and directly quantified in the results of the score-based classifier. The approach is evaluated on two datasets, showing to be robust to scarce and imbalanced data. Future work will focus on incorporating the time component into the approach, involving the application of the framework to videos and the creation of time series of the extracted attributes. Furthermore, the approach modeling potential will be explored on new use cases.

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
