# OpenReview forum: "An evidence-based neuro-symbolic framework for ambiguous image scene classification"
_nesyconf.org/NeSy/2025/Conference — NeSy 2025 Poster_

### Official Review · Reviewer_t9jP · 2025-03-24
**The work presents an interesting neuro-symbolic approach for ambiguous scenario classification. Although there is room for improvement, the paper is interesting and relevant to the research community.**

**Rating:** 6
**Confidence:** 3

**Review:**

The paper introduces a neuro-symbolic approach for handling ambiguous scenarios in image scene classification, composed of two main components. First, it leverages pre-trained deep learning models for object detection and depth estimation. Second, it employs multi-valued logic and evidence theory to improve interpretability and robustness, particularly in low-data contexts such as abandoned object detection. The approach is evaluated on two datasets, PETS2006 and AVS2007, showcasing encouraging results and illustrating its capability to handle uncertainty effectively.

Strengths:
- The methodology effectively combines deep learning and neuro-symbolic reasoning in a novel way.
- The use of modal logic and evidence theory to quantify uncertainty and ambiguity is clearly presented.
- Experiments demonstrate robustness against data imbalance and scarcity, showing practical advantages.

Weaknesses:
- Missing baseline comparisons.
- Lack of sensitivity analysis on pre-trained models.
- No formal evaluation of explainability.

Although the paper introduces a novel method based on a neuro-symbolic framework, there is room for improvement. Specifically, the paper lacks comparisons with baseline methods. While the authors acknowledge the dependency on pre-trained models, they do not quantify this dependency; an analysis using alternative models would help clarify this point. Additionally, although explainability is highlighted as a core strength, the manuscript currently lacks formal evaluation or visualization of this aspect. The authors also claim computational efficiency, but provide no empirical evidence in terms of runtime comparisons, inference time measurements, or complexity analysis. Nevertheless, the proposed approach yields interesting results and should be of interest to the community.

**Anonymity:**

Remain anonymous

---

### Official Review · Reviewer_ZfUa · 2025-04-06
**A hybrid of a neural perception with symbolic inference based on multi-valued mappings and elements of the Dempster-Shafer theory and modal logic**

**Rating:** 6
**Confidence:** 4

**Review:**

This paper combines a pair of neural network models (one for object detection and one for depth estimation) with a manually constructed rule-based inference pipeline that involves multi-valued mappings and elements of the Dempster-Shafer theory and modal logic to solve the task of abandoned object detection in images coming from monitoring/surveillance cameras.

While it is formally true that the proposed method is neurosymbolic by combining neural 'perception' with symbolic scene interpretation, the merger of these two components is not particularly deep. It essentially boils down to passing the detected objects, attirbuted with image features, to the rule-based inference engine. In particular, the system is not (and probably cannot be) trained as a whole, and the symbolic component is designed entirely by hand.

Unfortunately, the quality of presentation leaves something to be desired. In particular, the coverage of the formal underpinnings in Sec. 3 seems incoherent, which makes it really hard to understand the further line of argument. More specifically, in the second paragraph of that section:

> Given a subset A of X and a subset B of Y , ...

Why A? The subset A is not used in the following, which feels confusing.

Equation 1:
> Pl(B) = P (F−(B) | dom(F))
> Bel(B) = P (F +(B) |dom(F ))

For this notation to be coherent, dom(F) should be a logical predicate; but dom(F) is a set, as introduced a paragraph earlier.

> Pl and Bel are only well defined if P(dom(F)) > 0.

But again, what is the probability of *the set* dom(F)? How should one define such a notion?

On p. 8:

> □{abandoned}= min distance above 0.5 ∨ min distance above 0.75

Isn't the first condition on the right-hand side entailed by the second one?

Page 9: missing reference:
> As shown in Figure reffig:roc, both

Overall, the description of the theoretical concepts feels detached from the image-related processing introduced later in the paper. This could be addressed perhaps with some illustrative example, but sadly that wasn't provided.

The experimental lacks any baseline approach one could relate to and compare the proposed outcomes. I'm not even saying authors' approach needs to be necessarily better than the existing approaches, but without knowing what is the typical range of values of relevant metrics on the considered benchmarks, any discussion about the merits of the proposed approach becomes void.

In particular, it seems to me that the quite unique 'added value' of the proposed approach consists in supporting the inference with multi-valued mappings related to the Dempster-Shafer theory and modal logic. It becomes thus natural to ask what would be the impact of 'ablating' the proposed algorithm by devoiding it of the possibility of performing multi-valued mappings (by, perhaps, degenerating them to single-valued mappings). Unfortunately, it does not seem to me that this study offers such an analysis.

**Anonymity:**

Remain anonymous